# Approximating Real-Time Recurrent Learning with Random Kronecker Factors

**Asier Mujika** [*]
Department of Computer Science
ETH Zürich, Switzerland
asierm@inf.ethz.ch

**Florian Meier**
Department of Computer Science
ETH Zürich, Switzerland
meierflo@inf.ethz.ch

**Angelika Steger**
Department of Computer Science
ETH Zürich, Switzerland
steger@inf.ethz.ch

## Abstract

Despite all the impressive advances of recurrent neural networks, *sequential* data is still in need of better modelling. Truncated backpropagation through time (TBPTT), the learning algorithm most widely used in practice, suffers from the truncation bias, which drastically limits its ability to learn long-term dependencies.The Real-Time Recurrent Learning algorithm (RTRL) addresses this issue, but its high computational requirements make it infeasible in practice. The Unbiased Online Recurrent Optimization algorithm (UORO) approximates RTRL with a smaller runtime and memory cost, but with the disadvantage of obtaining noisy gradients that also limit its practical applicability. In this paper we propose the Kronecker Factored RTRL (KF-RTRL) algorithm that uses a Kronecker product decomposition to approximate the gradients for a large class of RNNs. We show that KF-RTRL is an unbiased and memory efficient online learning algorithm. Our theoretical analysis shows that, under reasonable assumptions, the noise introduced by our algorithm is not only stable over time but also asymptotically much smaller than the one of the UORO algorithm. We also confirm these theoretical results experimentally. Further, we show empirically that the KF-RTRL algorithm captures long-term dependencies and almost matches the performance of TBPTT on real world tasks by training Recurrent Highway Networks on a synthetic string memorization task and on the Penn TreeBank task, respectively. These results indicate that RTRL based approaches might be a promising future alternative to TBPTT.

## 1   Introduction

Processing sequential data is a central problem in the field of machine learning. In recent years, Recurrent Neural Networks (RNN) have achieved great success, outperforming all other approaches in many different sequential tasks like machine translation, language modeling, reinforcement learning and more.

Despite this success, it remains unclear how to train such models. The standard algorithm, Truncated Back Propagation Through Time (TBPTT) [18], considers the RNN as a feed-forward model over time with shared parameters. While this approach works extremely well in the range of a few hundred time-steps, it scales very poorly to longer time dependencies. As the time horizon is increased, the

---

[*]Author was supported by grant no. CRSII5_173721 of the Swiss National Science Foundation.

parameters are updated less frequently and more memory is required to store all past states. This makes TBPTT ill-suited for learning long-term dependencies in sequential tasks.

An appealing alternative to TBPTT is Real-Time Recurrent Learning (RTRL) [19]. This algorithm allows online updates of the parameters and learning arbitrarily long-term dependencies by exploiting the recurrent structure of the network for forward propagation of the gradient. Despite its impressive theoretical properties, RTRL is impractical for decently sized RNNs because run-time and memory costs scale poorly with network size.

As a remedy to this issue, Tallec and Ollivier [16] proposed the Unbiased Online Recurrent Learning algorithm (UORO). This algorithm unbiasedly approximates the gradients, which reduces the run-time and memory costs such that they are similar to the costs required to run the RNN forward. Unbiasedness is of central importance since it guarantees convergence to a local optimum. Still, the variance of the gradients slows down learning.

Here we propose the Kronecker Factored RTRL (KF-RTRL) algorithm. This algorithm builds up on the ideas of the UORO algorithm, but uses Kronecker factors for the RTRL approximation. We show both theoretically and empirically that this drastically reduces the noise in the approximation and greatly improves learning. However, this comes at the cost of requiring more computation and only being applicable to a class of RNNs. Still, this class of RNNs is very general and includes Tanh-RNN and Recurrent Highway Networks [20] among others.

The main contributions of this paper are:

- We propose the KF-RTRL online learning algorithm.
- We theoretically prove that our algorithm is unbiased and under reasonable assumptions the noise is stable over time and asymptotically by a factor $n$ smaller than the noise of UORO.
- We test KF-RTRL on a binary string memorization task where our networks can learn dependencies of up to 36 steps.
- We evaluate in character-level Penn TreeBank, where the performance of KF-RTRL almost matches the one of TBPTT for 25 steps.
- We empirically confirm that the variance of KF-RTRL is stable over time and that increasing the number of units does not increase the noise significantly.

## 2 Related Work

Training Recurrent Neural Networks for finite length sequences is currently almost exclusively done using BackPropagation Through Time [15] (BPTT). The network is "unrolled" over time and is considered as a feed-forward model with shared parameters (the same parameters are used at each time step). Like this, it is easy to do backpropagation and exactly calculate the gradients in order to do gradient descent.

However, this approach does not scale well to very long sequences, as the whole sequence needs to be processed before calculating the gradients, which makes training extremely slow and very memory intensive. In fact, BPTT cannot be applied to an online stream of data. In order to circumvent this issue, Truncated BackPropagation Through Time [18] (TBPTT) is used generally. The RNN is only "unrolled" for a fixed number of steps (the truncation horizon) and gradients beyond these steps are ignored. Therefore, if the truncation horizon is smaller than the length of the dependencies needed to solve a task, the network cannot learn it.

Several approaches have been proposed to deal with the truncation horizon. Anticipated Reweighted Truncated Backpropagation [17] samples different truncation horizons and weights the calculated gradients such that the expected gradient is that of the whole sequence. Jaderberg et al. [5] proposed Decoupled Neural Interfaces, where the network learns to predict incoming gradients from the future. Then, it uses these predictions for learning. The main assumption of this model is that all future gradients can be computed as a function of the current hidden state.

A more extreme proposal is calculating the gradients forward and not doing any kind of BPTT. This is known as Real-Time Recurrent Learning [19] (RTRL). RTRL allows updating the model parameters online after observing each input/output pair; we explain it in detail in Section 3. However, its large running time of order $O(n^4)$ and memory requirements of order $O(n^3)$, where $n$ is the number of

units of a fully connected RNN, make it unpractical for large networks. To fix this, Tallec and Ollivier [16] presented the Unbiased Online Recurrent Optimization (UORO) algorithm. This algorithm approximates RTRL using a low rank matrix. This makes the run-time of the algorithm of the same order as a single forward pass in an RNN, $O(n^2)$. However, the low rank approximation introduces a lot of variance, which negatively affects learning as we show in Section 5.

Other alternatives are Reservoir computing approaches [8] like Echo State Networks [6] or Liquid State Machines [9]. In these approaches, the recurrent weights are fixed and only the output connections are learned. This allows online learning, as gradients do not need to be propagated back in time. However, it prevents any kind of learning in the recurrent connections, which makes the RNN computationally much less powerful.

## 3   Real-Time Recurrent Learning and UORO

RTRL [19] is an online learning algorithm for RNNs. Contrary to TBPPT, no previous inputs or network states need to be stored. At any time-step $t$, RTRL only requires the hidden state $h_t$, input $x_t$ and $\frac{dh_{t-1}}{d\theta}$ in order to compute $\frac{dh_t}{d\theta}$. With $\frac{dh_t}{d\theta}$ at hand, $\frac{dL_t}{d\theta} = \frac{dL_t}{dh_t}\frac{dh_t}{d\theta}$ is obtained by applying the chain rule. Thus, the parameters can be updated online, that is, for each observed input/output pair one parameter update can be performed.

In order to present the RTRL update precisely, let us first define an RNN formally. An RNN is a differentiable function $f$, that maps an input $x_t$, a hidden state $h_{t-1}$ and parameters $\theta$ to the next hidden state $h_t = f(x_t, h_{t-1}, \theta)$. At any time-step $t$, RTRL computes $\frac{dh_t}{d\theta}$ by applying the chain rule:

$$\frac{dh_t}{d\theta} = \frac{\partial h_t}{\partial h_{t-1}}\frac{dh_{t-1}}{d\theta} + \frac{\partial h_t}{\partial x_t}\frac{dx_t}{d\theta} + \frac{\partial h_t}{\partial \theta} \tag{1}$$

$$= \frac{\partial h_t}{\partial h_{t-1}}\frac{dh_{t-1}}{d\theta} + \frac{\partial h_t}{\partial \theta} , \tag{2}$$

where the middle term vanishes because we assume that the inputs do not depend on the parameters. For notational simplicity, define $G_t := \frac{dh_t}{d\theta}$, $H_t := \frac{\partial h_t}{\partial h_{t-1}}$ and $F_t := \frac{\partial h_t}{\partial \theta}$, which reduces the above equation to

$$\frac{dh_t}{d\theta} = G_t = H_t G_{t-1} + F_t . \tag{3}$$

Both $F_t$ and $H_t$ are straight-forward to compute for RNNs. We assume $h_0$ to be fixed, which implies $G_0 = 0$. With all this, RTRL obtains the exact gradient $G_t$ for each time-step and enables online updates of the parameters. However, updating the parameters means that $G_t$ is only exact in the limit were the learning rate is arbitrarily small. This is because the $\theta$ that was used for computing $G_t$ is different from the $\theta$ after the parameter update. In practice learning rates are sufficiently small such that this is not an issue.

The downside of RTRL is that for a fully connected RNN with $n$ units the matrices $H_t$ and $G_t$ have size $n \times n$ and $n \times n^2$, respectively. Therefore, computing Equation 3 takes $\mathcal{O}(n^4)$ operations and requires $O(n^3)$ storage, which makes RTRL impractical for large networks.

The UORO algorithm [16] addresses this issue and reduces run-time and memory requirements to $O(n^2)$ at the cost of obtaining an unbiased but noisy estimate of $G_t$. More precisely, the UORO algorithm keeps an unbiased rank-one estimate of $G_t$ by approximating $G_t$ as the outer product $vw^T$ of two vectors of size $n$ and size $n^2$, respectively. At any time $t$, the UORO update consists of two approximation steps. Assume that the unbiased approximation $\hat{G}_{t-1} = vw^T$ of $G_{t-1}$ is given. First, $F_t$ is approximated by a rank-one matrix. Second, the sum of two rank-one matrices $H_t\hat{G}_{t-1} + F_t$ is approximated by a rank-one matrix yielding the estimate $\hat{G}_t$ of $G_t$. The estimate $\hat{G}_t$ is provably unbiased and the UORO update requires the same run-time and memory as updating the RNN [16].

# 4   Kronecker Factored RTRL

Our proposed Kronecker Factored RTRL algorithm (KF-RTRL) is an online learning algorithm for RNNs, which does not require storing any previous inputs or network states. KF-RTRL approximates $G_t$, which is the derivative of the internal state with respect to the parameters, see Section 3, by a Kronecker product. The following theorem shows that the KF-RTRL algorithm satisfies various desireable properties.

**Theorem 1.** *For the class of RNNs defined in Lemma 1, the estimate $G'_t$ obtained by the KF-RTRL algorithm satisfies*

1. *$G'_t$ is an unbiased estimate of $G_t$, that is $\mathbb{E}[G'_t] = G_t$, and*

2. *assuming that the spectral norm of $H_t$ is at most $1 - \epsilon$ for some arbitrary small $\epsilon > 0$, then at any time $t$, the mean of the variances of the entries of $G'_t$ is of order $O(n^{-1})$.*

*Moreover, one time-step of the KF-RTRL algorithm requires $O(n^3)$ operations and $O(n^2)$ memory.*

We remark that the class of RNNs defined in Lemma 1 contains many widely used RNN architectures like Recurrent Highway Networks and Tanh-RNNs, and can be extended to include LSTMs [4], see Section A.6. Further, the assumption that the spectral norm of $H_t$ is at most $1 - \epsilon$ is reasonable, as otherwise gradients might grow exponentially as noted by Bengio et al. [2]. Lastly, the bottleneck of the algorithm is a matrix multiplication and thus for sufficiently large matrices an algorithm with a better run time than $O(n^3)$ may be be practical.

In the remainder of this section, we explain the main ideas behind the KF-RTRL algorithm (formal proofs are given in the appendix). In the subsequent Section 5 we show that these theoretical properties carry over into practical application. KF-RTRL is well suited for learning long-term dependencies (see Section 5.1) and almost matches the performance of TBPTT on a complex real world task, that is, character level language modeling (see Section 5.2). Moreover, we confirm empirically that the variance of the KF-RTRL estimate is stable over time and scales well as the network size increases (see Section 5.3).

Before giving the theoretical background and motivating the key ideas of KF-RTRL, we give a brief overview of the KF-RTRL algorithm. At any time-step $t$, KF-RTRL maintains a vector $u_t$ and a matrix $A_t$, such that $G'_t = u_t \otimes A_t$ satisfies $\mathbb{E}[G'_t] = G_t$. Both $H_t G'_{t-1}$ and $F_t$ are factored as a Kronecker product, and then the sum of these two Kronecker products is approximated by one Kronecker product. This approximation step (see Lemma 2) works analogously to the second approximation step of the UORO algorithm (see rank-one trick, Proposition 1 in [16]). The detailed algorithmic steps of KF-RTRL are presented in Algorithm 1 and motivated below.

**Theoretical motivation of the KF-RTRL algorithm**

The key observation that motivates our algorithm is that for many popular RNN architectures $F$ can be exactly decomposed as the Kronecker product of a vector and a diagonal matrix, see Lemma 1. In the following Lemma, we show that this property is satisfied by a large class of RNNs that include many popular RNN architectures like Tanh-RNN and Recurrent Highway Networks. Intuitively, an RNN of this class computes several linear transformations (corresponding to the matrices $W^1, \ldots, W^r$) and merges the resulting vectors through pointwise non-linearities. For example, in the case of RHNs, there are two linear transformations that compute the new candidate cell and the forget gate, which then are merged through pointwise non-linearities to generate the new hidden state.

**Lemma 1.** *Assume the learnable parameters $\theta$ are a set of matrices $W^1, \ldots, W^r$, let $\hat{h}_{t-1}$ be the hidden state $h_{t-1}$ concatenated with the input $x_t$ and let $z^k = \hat{h}_{t-1} W^k$ for $k = 1, \ldots, r$. Assume that $h_t$ is obtained by point-wise operations over the $z^k$'s, that is, $(h_t)_j = f(z^1_j, \ldots, z^r_j)$, where $f$ is such that $\frac{\partial f}{\partial z^k_j}$ is bounded by a constant. Let $D^k \in \mathbb{R}^{n \times n}$ be the diagonal matrix defined by $D^k_{jj} = \frac{\partial (h_t)_j}{\partial z^k_j}$, and let $D = \left( D^1 | \ldots | D^r \right)$. Then, it holds $\frac{\partial h_t}{\partial \theta} = \hat{h}_{t-1} \otimes D$.*

Further, we observe that the sum of two Kronecker products can be approximated by a single Kronecker product in expectation. The following lemma, which is the analogue of Proposition 1 in [14] for Kronecker products, states how this is achieved.

---

**Algorithm 1** — One step of KF-RTRL (from time $t-1$ to $t$)

---

**Inputs:**
- input $x_t$, target $y_t$, previous recurrent state $h_{t-1}$ and parameters $\theta$
- $u_{t-1}$ and $A_{t-1}$ such that $\mathbb{E}\left[u_{t-1} \otimes A_{t-1}\right] = G_{t-1}$
- $SGDopt$ and $\eta_{t+1}$: stochastic optimizer and its learning rate

**Outputs:**
- new recurrent state $h_t$ and updated parameters $\theta$
- $u_t$ and $A_t$ such that $\mathbb{E}\left[u_t \otimes A_t\right] = G_t$

**/* Run one step of the RNN and compute the necessary matrices*/**

$z_j^k \leftarrow$ Compute linear transformations using $x_t$, $h_{t-1}$ and $\theta$

$h_t \leftarrow$ Compute $h_t$ using using point-wise operations over the $z_j^k$

$\hat{h}_{t-1} \leftarrow$ Concatenate $h_{t-1}$ and $x_t$

$D_{jj}^k \leftarrow \frac{\partial(h_t)_j}{\partial z_j^k} \qquad H \leftarrow \frac{\partial h_t}{\partial h_{t-1}} \qquad H' \leftarrow H \cdot A_{t-1}$

**/* Compute variance minimization and random multipliers */**

$p_1 \leftarrow \sqrt{\|H'\|_{HS}/\|u_{t-1}\|_{HS}} \qquad p_2 \leftarrow \sqrt{\|D\|_{HS}/\|\hat{h}_{t-1}\|_{HS}}$

$c_1, c_2 \leftarrow$ Independent random signs

**/* Compute next approximation */**

$u_t \leftarrow c_1 p_1 u_{t-1} + c_2 p_2 \hat{h}_{t-1} \qquad A_t \leftarrow c_1 \frac{1}{p_1} H' + c_2 \frac{1}{p_2} D$

**/* Compute gradients and update parameters */**

$L_{grad} \leftarrow u_t \otimes \left( \frac{\partial L(y_t, h_t)}{\partial h_t} \cdot A_t \right) \qquad SGDopt(L_{grad}, \eta_{t+1}, \theta)$

---

**Lemma 2.** *Let $C = A_1 \otimes B_1 + A_2 \otimes B_2$, where the matrix $A_1$ has the same size as the matrix $A_2$ and $B_1$ has the same size as $B_2$. Let $c_1$ and $c_2$ be chosen independently and uniformly at random from $\{-1, 1\}$ and let $p_1, p_2 > 0$ be positive reals. Define $A' = c_1 p_1 A_1 + c_2 p_2 A_2$ and $B' = c_1 \frac{1}{p_1} B_1 + c_2 \frac{1}{p_2} B_2$. Then, $A' \otimes B'$ is an unbiased approximation of $C$, that is $\mathbb{E}\left[A' \otimes B'\right] = C$. Moreover, the variance of this approximation is minimized by setting the free parameters $p_i = \sqrt{\|B_i\|/\|A_i\|}$.*

Lastly, we show by induction that random vectors $u_t$ and random matrices $A_t$ exist, such that $G_t' = u_t \otimes A_t$ satisfies $\mathbb{E}[G_t'] = G_t$. Assume that $G_{t-1}' = u_{t-1} \otimes A_{t-1}$ satisfies $\mathbb{E}[G_{t-1}'] = G_{t-1}$. Equation 3 and Lemma 1 imply that

$$G_t = H_t \mathbb{E}\left[G_{t-1}'\right] + F_t = H_t \mathbb{E}\left[u_{t-1} \otimes A_{t-1}\right] + \hat{h}_t \otimes D_t . \qquad (4)$$

Next, by linearity of expectation and since the first dimension of $u_{t-1}$ is 1, it follows

$$G_t = \mathbb{E}\left[H_t(u_{t-1} \otimes A_{t-1}) + \hat{h}_t \otimes D_t\right] = \mathbb{E}\left[u_{t-1} \otimes (H_t A_{t-1}) + \hat{h}_t \otimes D_t\right] . \qquad (5)$$

Finally, we obtain by Lemma 2 for any $p_1, p_2 > 0$

$$G_t = \mathbb{E}\left[(c_1 p_1 u_{t-1} + c_2 p_1 \hat{h}_t) \otimes (c_1 \frac{1}{p_1}(H_t A_{t-1}) + c_2 \frac{1}{p_2} D_t)\right] , \qquad (6)$$

where the expectation is taken over the probability distribution of $u_{t-1}$, $A_{t-1}$, $c_1$ and $c_2$.

With these observations at hand, we are ready to present the KF-RTRL algorithm. At any time-step $t$ we receive the estimates $u_{t-1}$ and $A_{t-1}$ from the previous time-step. First, compute $h_t$, $D_t$ and $H_t$. Then, choose $c_1$ and $c_2$ uniformly at random from $\{-1, +1\}$ and compute

$$u_t = c_1 p_1 u_{t-1} + c_2 p_2 \hat{h}_t \qquad (7)$$

$$A_t = c_1 \frac{1}{p_1}(H_t A_{t-1}) + c_2 \frac{1}{p_2} D_t , \qquad (8)$$

where $p_1 = \sqrt{\|H_t A_{t-1}\|/\|u_{t-1}\|}$ and $p_2 = \sqrt{\|D_t\|/\|\hat{h}_t\|}$. Lastly, our algorithm computes $\frac{dL_t}{dh_t} \cdot G_t'$, which is used for optimizing the parameters. For a detailed pseudo-code of the KF-RTRL algorithm

see Algorithm 1. In order to see that $\frac{dL_t}{dh_t} \cdot G'_t$ is an unbiased estimate of $\frac{dL_t}{d\theta}$, we apply once more linearity of expectation: $\mathbb{E}\left[\frac{dL_t}{dh_t} \cdot G'_t\right] = \frac{dL_t}{dh_t} \cdot \mathbb{E}\left[G'_t\right] = \frac{dL_t}{dh_t} \cdot G_t = \frac{dL_t}{d\theta}$.

One KF-RTRL update has run-time $O(n^3)$ and requires $O(n^2)$ memory. In order to see the statement for the memory requirement, note that all involved matrices and vectors have $O(n^2)$ elements, except $G'_t$. However, we do not need to explicitly compute $G'_t$ in order to obtain $\frac{dL_t}{d\theta}$, because $\frac{dL_t}{dh_t} \cdot G'_t = \frac{dL_t}{dh_t} \cdot u_t \otimes A_t = u_t \otimes (\frac{dL_t}{dh_t} A_t)$ can be evaluated in this order. In order to see the statement for the run-time, note that $H_t$ and $A_{t-1}$ have both size $O(n) \times O(n)$. Therefore, computing $H_t A_{t-1}$ requires $O(n^3)$ operations. All other arithmetic operations trivially require run-time $O(n^2)$.

The proofs of Lemmas 1 and 2 and of the second statement of Theorem 1 are given in the appendix.

**Comparison of the KF-RTRL with the UORO algorithm**

Since the variance of the gradient estimate is directly linked to convergence speed and performance, let us first compare the variance of the two algorithms. Theorem 1 states that the mean of the variances of the entries of $G'_t$ are of order $O(n^{-1})$. In the appendix, we show a slightly stronger statement, that is, if $\|F_t\| \leq C$ for all $t$, then the mean of the variances of the entries is of order $O(\frac{C^2}{n^3})$, where $n^3$ is the number of elements of $G_t$. The bound $O(n^{-1})$ is obtained by a trivial bound on the size of the entries of $h_t$ and $D_t$ and using $\|h_t\|\|D_t\| = \|F_t\|$. In the appendix, we show further that already the first step of the UORO approximation, in which $F_t$ is approximated by a rank-one matrix, introduces noise of order $(n-1)\|F_t\|$. Assuming that all further approximations would not add any noise, then the same trivial bounds on $\|F_t\|$ yield a mean variance of $O(1)$. We conclude that the variance of KF-RTRL is asymptotically by (at least) a factor $n$ smaller than the variance of UORO.

Next, let us compare the generality of the algorithm when applying it to different network architectures. The KF-RTRL algorithms requires that in one time-step each parameter only affects one element of the next hidden state (see Lemma 1). Although many widely used RNN architectures satisfy this requirement, seen from this angle, the UORO algorithm is favorable as it is applicable to arbitrary RNN architectures.

Finally, let us compare memory requirements and runtime of KF-RTRL and UORO. In terms of memory requirements, both algorithms require $O(n^2)$ storage and perform equally good. In terms of run-time, KF-RTRL requires $O(n^3)$, while UORO requires $O(n^2)$ operations. However, the faster run-time of UORO comes at the cost of worse variance and therefore worse performance. In order to reduce the variance of UORO by a factor $n$, one would need $n$ independent samples of $G'_t$. This could be achieved by reducing the learning rate by a factor of $n$, which would then require $n$ times more data, or by sampling $G'_t$ $n$ times in parallel, which would require $n$ times more memory. Still, our empirical investigation shows, that KF-RTRL outperforms UORO, even when taking $n$ UORO samples of $G_t$ to reduce the variance (see Figure 3). Moreover, for sufficiently large networks the scaling of the KF-RTRL run-time improves by using fast matrix multiplication algorithms.

# 5 Experiments

In this section, we quantify the effect on learning that the reduced variance of KF-RTRL compared to the one of UORO has. First, we evaluate the ability of learning long-term dependencies on a deterministic binary string memorization task. Since most real world problems are more complex and of stochastic nature, we secondly evaluate the performance of the learning algorithms on a character level language modeling task, which is a more realistic benchmark. For these two tasks, we also compare to learning with Truncated BPTT and measure the performance of the considered algorithms based on 'data time', i.e. the amount of data observed by the algorithm. Finally, we investigate the variance of KF-RTRL and UORO by comparing to their exact counterpart, RTRL. For all experiments, we use a single-layer Recurrent Highway Network [20] [2].

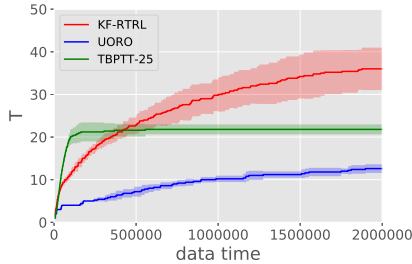

Input: #01101------
Output: ------#01101

Input: #11010------
Output: ------#11010

Input: #00100------
Output: ------#00100

(a)                                                    (b)

**Figure 1:** Copy Task: Figure 1(a) shows the learning curves of UORO, KF-RTRL and TBPTT with truncation horizon of 25 steps. We plot the mean and standard deviation (shaded area) over 5 trials. Figure 1(b) shows three input and output examples with $T = 5$.

## 5.1 Copy Task

In the copy task experiment, a binary string is presented sequentially to an RNN. Once the full string has been presented, it should reconstruct the original string without any further information. Figure 1(b) shows several input-output pairs. We refer to the length of the string as $T$. Figure 1(a) summarizes the results. The smaller variance of KF-RTRL greatly helps learning faster and capturing longer dependencies. KF-RTRL and UORO manage to solve the task on average up to $T = 36$ and $T = 13$, respectively. As expected, TBPTT cannot learn dependencies that are longer than the truncation horizon.

In this experiment, we start with $T = 1$ and when the RNN error drops below $0.15$ bits/char, we increase $T$ by one. After each sequence, the hidden state is reset to all zeros. To improve performance, the length of each sample is picked uniformly at random from $T$ to $T - 5$. This forces the network to learn a general algorithm for the task, rather than just learning to solve sequences of length $T$. We use a RHN with 256 units and a batch size of 256. We optimize the log-likelihood using the Adam optimizer [7] with default Tensorflow [1] parameters, $\beta_1 = 0.9$ and $\beta_2 = 0.999$. For each model we pick the optimal learning rate from $\{10^{-2.5}, 10^{-3}, 10^{-3.5}, 10^{-4}\}$. We repeat each experiment 5 times.

## 5.2 Character level language modeling on the Penn Treebank dataset

A standard test for RNNs is character level language modeling. The network receives a text sequentially, character by character, and at each time-step it must predict the next character. This task is very challenging, as it requires both long and short term dependencies. Additionally, it is highly stochastic, i.e. for the same input sequence there are many possible continuations, but only one is observed at each training step. Figure 2 and Table 1 summarize the results. Truncated BPTT outperforms both online learning algorithms, but KF-RTRL almost reaches its performance and considerably outperforms UORO. For this task, the noise introduced in the approximation is more harmful than the truncation bias. This is probably the case because the short term dependencies dominate the loss, as indicated by the small difference between TBPTT with truncation horizon 5 and 25.

For this experiment we use the Penn TreeBank [10] dataset, which is a collection of Wall Street Journal articles. The text is lower cased and the vocabulary is restricted to 10K words. Out of vocabulary words are replaced by "<unk>" and numbers by "N". We split the data following Mikolov et al. [13]. The experimental setup is the same as in the Copy task, and we pick the optimal learning rate from the same range. Apart from that, we reset the hidden state to the all zeros state with probability $0.01$ at each time step. This technique was introduced by Melis et al. [11] to improve the performance on the validation set, as the initial state for the validation is the all zeros state. Additionally, this helps the online learning algorithms, as it resets the gradient approximation, getting rid of stale gradients. Similar techniques have been shown [3] to also improve RTRL.

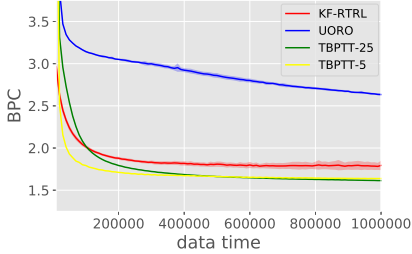

**Figure 2:** Validation performance on Penn TreeBank in bits per character (BPC). The small variance of the KF-RTRL approximation considerably improves the performance compared to UORO.

**Table 1:** Results on Penn TreeBank. Merity et al. [12] is currently the state of the art (trained with TBPTT). For simplicity we do not report standard deviations, as all of them are smaller than 0.03.

| Name | Validation | Test | #params |
|---|---|---|---|
| KF-RTRL | 1.77 | 1.72 | 133K |
| UORO | 2.63 | 2.61 | 133K |
| TBPTT-5 | 1.64 | 1.58 | 133K |
| TBPTT-25 | 1.61 | 1.56 | 133K |
| Merity et al. [12] | - | 1.18 | 13.8M |

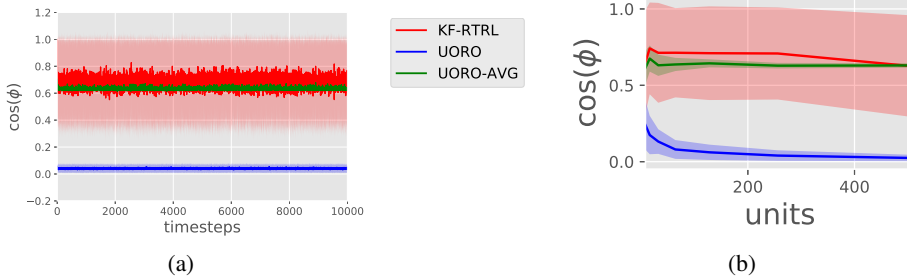

(a)  (b)

**Figure 3:** Variance analysis: We compare the cosine of the angle between the approximated and the true value of $\frac{dL}{d\theta}$. A cosine of 1 implies that the approximation and the true value are exactly aligned, while a random vector gets a cosine of 0 in expectation. Figure 3(a) shows that the variance is stable over time for the three algorithms. Figure 3(b) shows that the variance of KF-RTRL and UORO-AVG are almost unaffected by the number of units, while UORO degrades more quickly as the network size increases.

## 5.3 Variance Analysis

With our final set of experiments, we empirically measure how the noise evolves over time and how it is affected by the number of units $n$. Here, we also compare to UORO-AVG that computes $n$ independent samples of UORO and averages them to reduce the variance. The computation costs of UORO-AVG are on par with those of KF-RTRL, $O(n^3)$, however the memory costs of $O(n^3)$ are higher than the ones of KF-RTRL of $O(n^2)$. For each experiment, we compute the angle $\phi$ between the gradient estimate and the exact gradient of the loss with respect to the parameters. Intuitively, $\phi$ measures how aligned the gradients are, even if the magnitude is different. Figure 3(a) shows that $\phi$ is stable over time and the noise does not accumulate for any of the three algorithms. Figure 3(b) shows that KF-RTRL and UORO-AVG have similar performance as the number of units increases. This observation is in line with the theoretical prediction in Section A.5 that the variance of UORO is by a factor $n$ larger than the KF-RTRL variance (averaging $n$ samples as done in AVG-UORO reduces the variance by a factor $n$).

In the first experiment, we run several untrained RHNs with 256 units over the first 10000 characters of Penn TreeBank. In the second experiment, we compute $\phi$ after running RHNs with different number of units for 100 steps on Penn TreeBank. We perform 100 repetitions per experiment and plot the mean and standard deviation.

## 6 Conclusion

In this paper, we have presented the KF-RTRL online learning algorithm. We have proven that it approximates RTRL in an unbiased way, and that under reasonable assumptions the noise is stable over time and much smaller than the one of UORO, the only other previously known unbiased RTRL

approximation algorithm. Additionally, we have empirically verified that the reduced variance of our algorithm greatly improves learning for the two tested tasks. In the first task, an RHN trained with KF-RTRL effectively captures long-term dependencies (it learns to memorize binary strings of length up to 36). In the second task, it almost matches the performance of TBPTT in a standard RNN benchmark, character level language modeling on Penn TreeBank.

More importantly, our work opens up interesting directions for future work, as even minor reductions of the noise could make the approach a viable alternative to TBPTT, especially for tasks with inherent long-term dependencies. For example constraining the weights, constraining the activations or using some form of regularization could reduce the noise. Further, it may be possible to design architectures that make the approximation less noisy. Moreover, one might attempt to improve the run-time of KF-RTRL by using approximate matrix multiplication algorithms or inducing properties on the $H_t$ matrix that allow for fast matrix multiplications, like sparsity or low-rank.

This work advances the understanding of how unbiased gradients can be computed, which is of central importance as unbiasedness is essential for theoretical convergence guarantees. Since RTRL based approaches satisfy this key assumption, it is of interest to further progress them.

## Footnotes

[2] For implementation simplicity, we use $2 * sigmoid(x) - 1$ instead of $Tanh(x)$ as non-linearity function. Both functions have very similar properties, and therefore, we do not believe this has any significant effect.

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
