[Supplementary Material]

# A  Appendix

In this appendix, we prove all the lemmas and theorems whose proofs has been omitted in the main paper. For the ease of readability we restate the statement object for proving in the beginning of each section.

## A.1  Basic Notation

The Hilbert-Schmid norm of a matrix $A$ is defined as $\|A\|_{HS} := \sum_{i,j} a_{ij}^2$ and the Hilbert-Schmid inner product of two matrices $A, B$ of the same size is defined as $\langle A, B \rangle_{HS} = \sum_{ij} a_{ij} b_{ij}$. When regarding an $n \times m$ matrix as a point in $\mathbb{R}^{mn}$, then the standard euclidian norm of this point is the same as the Hilbert-Schmid norm of the matrix. Therefore, for notational simplicity, we omit the $HS$ subscript and write $\|A\|$ and $\langle A, B \rangle$. Note that the Hilbert-Schmid norm satisfies $\|A \otimes B\| = \|A\|\|B\|$. Further, we measure the variance of a random matrix $A$ by the sum of the variances of its entries:

$$\mathrm{Var}[A] = \sum_{ij} \mathrm{Var}[a_{ij}] = \sum_{ij} \mathbb{E}[a_{ij}^2] - \mathbb{E}[a_{ij}]^2 = \mathbb{E}[\|A\|^2] - \|\mathbb{E}[A]\|^2 \tag{9}$$

## A.2  Proof of Lemma 1

**Lemma.** *Assume the learnable parameters $\theta$ are a set of matrices $W^1, \ldots, W^r$, let $\hat{h}_{t-1}$ be the hidden state $h_{t-1}$ concatenated with the input $x_t$ and let $z^k = \hat{h}_{t-1} W^k$ for $k = 1, \ldots, r$. Assume that $h_t$ is obtained by point-wise operations over the $z^k$'s, that is, $(h_t)_j = f(z_j^1, \ldots, z_j^r)$, where $f$ is such that $\frac{\partial f}{\partial z_j^k}$ is bounded by a constant. Let $D^k \in \mathbb{R}^{n \times n}$ be the diagonal matrix defined by $D_{jj}^k = \frac{\partial (h_t)_j}{\partial z_j^k}$, and let $D = \left( D^1 | \ldots | D^r \right)$. Then, it holds $\frac{\partial h_t}{\partial \theta} = \hat{h}_{t-1} \otimes D$.*

*Proof.* Note that $z_a^b = \sum_i w_{ia}^b (\hat{h}_{t-1})_i$ only depends on $w_{ij}^k$ if $j = a$ and $k = b$, that $\frac{\partial z_j^k}{\partial w_{ij}^k} = (\hat{h}_{t-1})_i$, and that $\frac{\partial (h_t)_\ell}{\partial z_j^i} = 0$ if $j \neq \ell$. Therefore

$$\frac{\partial (h_t)_\ell}{\partial w_{ij}^k} = \sum_{a,b} \frac{\partial (h_t)_\ell}{\partial z_a^b} \frac{\partial z_a^b}{\partial w_{ij}^k} = \frac{\partial (h_t)_\ell}{\partial z_\ell^k} \frac{\partial z_\ell^k}{\partial w_{ij}^k} = \frac{\partial (h_t)_\ell}{\partial z_\ell^k} \cdot \delta_{\ell,j} (\hat{h}_{t-1})_i \,, \tag{10}$$

where $\delta_{\ell,j}$ is the Kronecker delta, which is 1 if $\ell = j$ and 0 if $\ell \neq j$. If we assume that the parameters $w_{ij}^k$ are ordered lexicographically in $i$, $k$, $j$, then $D_{\ell,j}^k = \delta_{\ell,j} \frac{\partial (h_t)_j}{\partial z_j^k}$. $\square$

## A.3  Proof of Lemma 2

As mentioned in the paper Lemma 2 is essentially borrowed from [14]. We state the lemma slightly more general as in the paper, that is, for arbitrary many summands.

**Lemma 3.** *Let $C = \sum_{i=1}^m A_i \otimes B_i$, where the $A_i$'s are of the same size and the $B_i'$s are of the same size. Let the $c_1, \ldots, c_m$ be chosen independently and uniformly at random from $\{-1, +1\}$ and let $p_1, \ldots, p_m > 0$ be positive reals. Define $A' = \sum_{i=1}^m c_i p_i A_i$ and $B' = \sum_{i=1}^m c_i \frac{1}{p_i} B_i$. Then, $C' = A' \otimes B'$ is an unbiased approximation of $C$, that is $\mathbb{E}\left[ C' \right] = C$. The free parameters $p_i$ can be chosen to minimize the variance of $A'$. For the optimal choice $p_i = \sqrt{\|B_i\|/\|v_i\|}$ it holds*

$$Var[C'] = \sum_i \sum_{i \neq j} \|A_i\|\|A_j\|\|B_i\|\|B_j\| + \langle A_i, A_j \rangle \langle B_i, B_j \rangle \,. \tag{11}$$

*Proof.* The independence of the $c_i$ implies that $\mathbb{E}[c_i c_j] = 1$ if $i = j$ and $\mathbb{E}[c_i c_j] = 0$ if $i \neq j$. Therefore, the first claim follows easily by linearity of expectation:

$$\mathbb{E}[C'] = \mathbb{E}[(\sum_i c_i p_i A_i) \otimes (\sum_j c_j \frac{1}{p_j} B_j)] = \sum_i \sum_j \mathbb{E}[c_i c_j] \frac{p_i}{p_j} A_i \otimes B_j = \sum_i A_i \otimes B_i \,.$$

For the proof of the second claim we use Proposition 1 from [14]. Let $D = \sum_i \vec{A}_i \vec{B}_i^T$, where $\vec{A}$ denotes the vector obtained by concatenating the columns of a matrix $A$, and let $D' = (\sum_i c_i p_i \vec{A}_i)(\sum_j c_j \frac{1}{p_j} \vec{B}_j)^T$. $C'$ and $D'$ have same entries for the same choice of $c_i$'s. It follows that $\|\mathbb{E}[C']\| = \|\mathbb{E}[D']\|$, $\mathbb{E}[\|C'\|^2] = \mathbb{E}[\|D'\|^2]$, and therefore $\text{Var}[C'] = \text{Var}[D']$. By Proposition 1 from [14], choosing $p_i = \sqrt{\|B_i\|/\|A_i\|}$ minimizes $\text{Var}[D']$ resulting in $\text{Var}[D'] = \sum_i \sum_{i \neq j} \|\vec{A}_i\| \|\vec{A}_j\| \|\vec{B}_i\| \|\vec{B}_j\| + \langle \vec{A}_i, \vec{A}_j \rangle \langle \vec{B}_i, \vec{B}_j \rangle$. This implies the Lemma because $\|A\| = \|\vec{A}_i\|$ and $\langle A, B \rangle = \langle \vec{A}, \vec{B} \rangle$ for any matrices $A$ and $B$. $\qquad \square$

## A.4   Proof of Theorem 1

The spectral norm for matrix A is defined as $\sigma(A) := \max_{v:\|v\|=1} \|Av\|$. Note that $\|AB\| \leq \sigma(A)\|B\|$ holds for any matrix $B$.

**Theorem 2.** *Let $\epsilon > 0$ be arbitrary small. Assume for all $t$ that the spectral norm of $H_t$ is at most $1 - \epsilon$, $\|\hat{h}_t\| \leq C_1$ and $\|D_t\| \leq C_2$. Then for the class of RNNs defined in Lemma 1, the estimate $G_t'$ obtained by the KF-RTRL algorithm satisfies at any time $t$ that $\text{Var}[G_t'] \leq \frac{16}{\epsilon^3(2-\epsilon)} C_1^2 C_2^2$.*

Before proving this theorem let us show how it implies Theorem 1. Note that the hidden state $h_t$ and the inputs $x_t$ take values between $-1$ and $1$. Therefore, $\|\hat{h}_t\|^2 = O(n)$. By Lemma 1 the $rn$ non-zero entries of $D$ are of the form $\frac{\partial (h_t)_j}{\partial z_j^k} = \frac{\partial f}{\partial z_j^k}$. By the assumptions on $f$ the entries of $D_t$ are bounded and $\|D_t\|^2 = O(n)$ follows. Theorem 2 implies that $\text{Var}[G_t'] = O(n^2)$. Since the number of entries in $G_t'$ is of order $\Theta(n^3)$, the mean of the variances of the entries of $G_t'$ is of order $O(n^{-1})$.

*Proof of Theorem 2.* The proof idea goes as follows. Write $G_t' = G_t + \hat{G}_t$ as the sum of the true (deterministic) value $G_t = \frac{dh_t}{d\theta}$ of the gradient and the random noise $\hat{G}_t$ induced by the approximations until time $t$. Note that $\text{Var}[G_t'] = \text{Var}[\hat{G}_t]$. Then, write $\text{Var}[\hat{G}_t]$ as the sum of the variance induced by the $t$-th time step and the variance induced by previous steps. The bound on the spectral norm of $H_t$ ensures that the latter summand can be bounded by $(1-\epsilon)^2 \text{Var}[\hat{G}_{t-1}]$. Therefore the variance stays of the same order of magnitude as the one induced in each time-step and this magnitude can be bounded as well.

Now let us prove the statement formally. Define

$$B := \frac{p_1}{p_2} u_{t-1} \otimes D_t + \frac{p_2}{p_1} \hat{h}_t \otimes H_t A_{t-1} \qquad (12)$$

By equation Equation 7 and 8

$$G_t' = (p_1 c_1 u_{t-1} + p_2 c_2 \hat{h}_t) \otimes (\tfrac{c_1}{p_1} H_t A_{t-1} + \tfrac{c_2}{p_2} D_t) \qquad (13)$$

$$= u_{t-1} \otimes H_t A_{t-1} + \hat{h}_t \otimes D_t + c_1 c_2 B \qquad (14)$$

Observe that

$$u_{t-1} \otimes H_t A_{t-1} = H_t(u_{t-1} \otimes A_{t-1}) = H_t G_{t-1}' = H_t G_{t-1} + H_t \hat{G}_{t-1} , \qquad (15)$$

which implies together with Equation 3 that

$$G_t + \hat{G}_t = G_t' = H_t G_{t-1} + H_t \hat{G}_{t-1} + \hat{h}_t \otimes D_t + c_1 c_2 B = G_t + H_t \hat{G}_{t-1} + c_1 c_2 B . \qquad (16)$$

It follows that $\hat{G}_t = H_t \hat{G}_{t-1} + c_1 c_2 B$.

**Claim 1.** *For two random matrices $A$ and $B$ and $c$ chosen uniformly at random in $\{-1, +1\}$ independent from $A$ and $B$, it holds $\text{Var}[A + cB] = \text{Var}[A] + \mathbb{E}[\|B\|^2]$.*

We postpone the proof and first show the theorem. Claim 1 implies that

$$\text{Var}[\hat{G}_t] = \text{Var}[H_t \hat{G}_{t-1}] + \mathbb{E}[\|B\|^2] . \qquad (17)$$

Let us first bound the first term. Since $G'_t$ is unbiased, it holds $\mathbb{E}[\hat{G}_{t-1}] = 0$, $\text{Var}[\hat{G}_{t-1}] = \mathbb{E}[\|\hat{G}_{t-1}\|^2]$, and therefore

$$\text{Var}[H_t\hat{G}_{t-1}] = \mathbb{E}[\|H_t\hat{G}_{t-1}\|^2] - \|\mathbb{E}[H_t\hat{G}_{t-1}]\|^2$$
$$\leq (1-\epsilon)^2 \mathbb{E}[\|\hat{G}_{t-1}\|^2]$$
$$= (1-\epsilon)^2 \text{Var}[\hat{G}_{t-1}] .$$

A bound for the second term can be obtained by the triangle inequality:

$$\|B\| \leq \|\frac{p_1}{p_2}u_{t-1} \otimes D_t\| + \|\frac{p_2}{p_1}\hat{h}_t \otimes H_tA_{t-1}\| \tag{18}$$
$$= 2\left(\|u_t\|\|\hat{h}_t\|\|D_t\|\|H_tA_{t-1}\|\right)^{1/2} \tag{19}$$
$$\leq \frac{4}{\epsilon}C_1C_2 , \tag{20}$$

where the last inequality follows the following claim.

**Claim 2.** $\|u_t\|\|A_t\| \leq \frac{4C_1C_2}{\epsilon^2}$ *holds for all time-step $t$.*

Let us postpone the proof and show by induction that $\text{Var}[\hat{G}_t] \leq \frac{16}{\epsilon^3(2-\epsilon)}C_1^2C_2^2$. Assume this is true for $t-1$, then

$$\text{Var}[\hat{G}_t] = \text{Var}[H_t\hat{G}_{t-1}] + \mathbb{E}[\|B\|^2] \tag{21}$$
$$\leq (1-\epsilon)^2\text{Var}[\hat{G}_{t-1}] + (\frac{4}{\epsilon}C_1C_2)^2 \tag{22}$$
$$\leq (1-\epsilon)^2\frac{16}{\epsilon^3(2-\epsilon)}C_1^2C_2^2 + (\frac{4}{\epsilon}C_1C_2)^2 \tag{23}$$
$$= \frac{16}{\epsilon^3(2-\epsilon)}C_1^2C_2^2 , \tag{24}$$

which implies the theorem. Let us first prove Claim 1. Note that $\mathbb{E}[cX] = 0$ holds for any random variable $X$, and therefore

$$\text{Var}[A + cB] = \sum_{ij}\text{Var}[A_{ij} + cB_{ij}] \tag{25}$$
$$= \sum_{ij}\mathbb{E}[(A_{ij} + cB_{ij})^2] - \mathbb{E}[A_{ij} + cB_{ij}]^2 \tag{26}$$
$$= \sum_{ij}\mathbb{E}[A_{ij}^2] - \mathbb{E}[A_{ij}]^2 + \mathbb{E}[c^2B_{ij}^2] \tag{27}$$
$$= \text{Var}[A] + \mathbb{E}[\|B\|^2] . \tag{28}$$

It remains to prove Claim 2. $\qquad\qquad\square$

We show this claim by induction over $t$. For $t = 0$ this is true since $G_0$ is the all zero matrix. For the induction step let us assume that $\|u_{t-1}\|\|A_{t-1}\| \leq \frac{4C_1C_2}{\epsilon^2}$. Using our update rules for $u_t$ and $A_t$ (see Equation 7 and 8) and the triangle inequality we obtain $\|u_t\| \leq \sqrt{\|H_tA_{t-1}\|\|u_{t-1}\|} + \sqrt{\|\hat{h}_t\|\|D_t\|}$

and $\|A_t\| \leq \sqrt{\|H_t A_{t-1}\| \|u_{t-1}\|} + \sqrt{\|\hat{h}_t\| \|D_t\|}$. It follows that

$$\|u_t\| \|A_t\| \leq \left( \sqrt{\|H_t A_{t-1}\| \|u_{t-1}\|} + \sqrt{\|\hat{h}_t\| \|D_t\|} \right)^2 \tag{29}$$

$$\leq \left( \sqrt{(1-\epsilon)\|A_{t-1}\| \|u_{t-1}\|} + \sqrt{\|\hat{h}_t\| \|D_t\|} \right)^2 \tag{30}$$

$$\leq \left( \sqrt{(1-\epsilon)\frac{4C_1 C_2}{\epsilon^2}} + \sqrt{C_1 C_2} \right)^2 \tag{31}$$

$$= \left( \sqrt{(1-\epsilon)} \cdot \tfrac{2}{\epsilon} + 1 \right)^2 C_1 C_2 \tag{32}$$

$$\leq \left( \sqrt{(1-\epsilon+\tfrac{\epsilon^2}{4})} \cdot \tfrac{2}{\epsilon} + 1 \right)^2 C_1 C_2 \tag{33}$$

$$= \left( (1-\tfrac{\epsilon}{2})\tfrac{2}{\epsilon} + 1 \right)^2 C_1 C_2 \tag{34}$$

$$= \frac{4C_1 C_2}{\epsilon^2} \ . \tag{35}$$

## A.5 Computation of Variance of UORO Approach

In the first approximation step of the UORO algorithm $F_t$ is approximated by a rank one matrix $vv^T F_t$, where $v$ is chosen uniformly at random from $\{-1, +1\}^n$. For the RNN architectures considered in this paper, $F_t$ is a concatenation of diagonal matrices, cf. Lemma 1. Intuitively, all the non-diagonal elements of the UORO approximation are far off the true value $0$. Therefore, the variance per entry introduced in this step will be of order of the diagonal entries of $F_t$ . More precisely, it holds that

$$\mathrm{Var}[vv^T F_t] = \sum_{i,j} \mathbb{E}[(v_i v_j (F_t)_{jj})^2] - \mathbb{E}[v_i v_j (F_t)_{jj}]^2 = \sum_{i \neq j} (F_t)_{jj}^2 = (n-1)\|F_t\|^2 \ , \tag{36}$$

where we used that $F_t$ is diagonal, $\mathbb{E}[v_i v_j] = 0$ if $i \neq j$ and $\mathbb{E}[v_i v_j] = 1$ if $i = j$.

Recall that for $h_t$ and $D_t$ of the KF-RTRL algorithm, it holds $\|h_t\| \|D_t\| = \|F_t\|$ and that we bounded the variance of the gradient estimate essentially by $\|h_t\|^2 \|D_t\|^2 = \|F_t\|^2$ (actually, we bounded it by using an upper bound $C_1 C_2$ of $\|h\| \|D\|$). Therefore, the first approximation step of one UORO update introduces a variance that is by a factor $n$ larger than the total variance of the KF-RTRL approximation. Assuming that the entries of $F_t$ are of constant size (as assumed for obtaining the $O(n^{-1})$ bound per entry for KF-RTRL), implies this first UORO approximation step has constant variance per entry. The second UORO approximation step can only increase the variance. We remark that with the same assumption on the spectral norm as in Theorem 2, one could similarly derive a bound of $O(1)$ on the mean variance per entry of the UORO algorithm.

## A.6 Extending KF-RTRL to LSTMs

Our approach can also be applied to LSTMs. However, it requires more computation and memory because an LSTM has twice as many parameters as an RHN for the same number of units. Additionally, the hidden state is twice as large as it consists of a cell state, $c_t$, and a hidden state, $h_t$. To apply KF-RTRL to LSTMs, we have to show that $\frac{\partial (c_t | h_t)}{\partial \theta}$ can be exactly decomposed as a Kronecker product of the same form as in Lemma 1. If we show this, then the rest of the algorithm can easily be applied. The following equations define the transition function of an LSTM:

$$\begin{pmatrix} f_t \\ i_t \\ o_t \\ g_t \end{pmatrix} = W_h h_{t-1} + W_x x_t$$
$$c_t = \sigma(f_t) \odot c_{t-1} + \sigma(i_t) \odot \tanh(g_t)$$
$$h_t = \sigma(o_t) \odot \tanh(c_t)$$

Observe that both $c_t$ and $h_t$ are the results of linear operations followed by point-wise operations, as is required to apply Lemma 1. Thus, we get that $\frac{\partial c_t}{\partial \theta} = h'_{t-1} \otimes D_{c_t}$ and $\frac{\partial h_t}{\partial \theta} = h'_{t-1} \otimes D_{h_t}$, where $h'_{t-1}$ is the concatenation of the input and hidden states, as in Lemma 1. Consequently $\frac{\partial (c_t | h_t)}{\partial \theta} = h'_{t-1} \otimes \begin{pmatrix} D_{c_t} \\ D_{h_t} \end{pmatrix}$.