[Reviews · NeurIPS 2018]

Reviewer 1



Summary: This paper investigates Real-Time Recurrent Learning (RTRL) for training recurrent neural networks. As RTRL has a high-computational/memory costs, authors propose to approximate RTRL using the Kronecker Factorization. For a subclass of RNN, authors demonstrate that their approximation, KF-RTRL, is an unbiased estimate of the true gradient and has lower variance than previously proposed RTRL approximation such as UORO (relying on a low-rank approximation). KF-RTRL is empirically evaluated on a copy and language modeling task. Authors validate that KF-RTRL is a competitive alternative to UORO, also it underperforms truncated Backpropagation through time. Clarity: The paper is well written and pleasant to read. Few minors point: - While z is properly defined in the text, it is not clear what z is from algorithm 1 alone. - What does the x-axis correspond to in Figure (1) (a) and Figure (2) ? Is it the overall training time in second? Quality: The submission seems technically sound. KF-RTRL shows an advantage over UORO both theoretically and in the empirical evaluation. Authors also compare their approach with truncated backpropagation through time, and show that RTRL approaches can still be improved. It would be informative to reports the memory requirement/running time of the different algorithms. In particular, does KF-RTRL outperform UORO in term of training time ? Is TBPTT-25 more memory intensive than KF-RTRL? It seems that KF-RTRL do not capture well long-term dependency? The maximum T used in the copy experiment is 36 while in the second experiment TBPTT with a context of 25 outperforms KF-RTRL. Is it related to the requirement on the spectral norm in Theorem 1., which states that the spectral norm on H_t must be less than 1, hence leading to vanishing gradients? Originality: Using Kronecker Factorization to approximate RTRL algorithm appears novel to me. Significance. The memory cost associated with training RNN is an important limitation. Tackling this issue could allow the exploration of larger RNN models. This paper is one step in this direction and seems relevant to the community. This paper introduces a novel approximation of RTRL which improve upon previous RTRL method. While KF-RTRL does not outperform truncated backpropagation through time, it is one step going in that direction. The paper is well written and pleasant to read, and the proposed method is interesting. For those reasons, I recommend acceptance. After Rebuttal: Thanks for the feedback! After reading the authors respond and the other reviews, I still feel that score of 7 is appropriate for this paper.

Reviewer 2



SUMMARY: This paper proposes a Kronecker factored variant of the Real-Time Recurrent Learning algorithm (RTRL) proposed by Ollivier et al, called KF-RTRL. The main obstacle of using RTRL in place of Truncated Back-Propagation Through Time (BPTT) to train RNN is its computational complexity of O(n^4) where n is the number of units. To address this issue, the Unbiased Online Recurrent Optimization (UORO) algorithm proposed to approximate RTRL using a rank one matrix to approximate the matrix of derivatives of the hidden state h_t wrt the parameters (henceforth denoted by G_t, of size n x n^2), reducing the complexity to O(n^2) at the cost of introducing variance to the unbiased estimator of the gradient. Instead, KF-RTRL approximates G_t by the Kronecker product of a 1xn vector and an nxn matrix, to obtain an unbiased estimator with less variance with a complexity of O(n^3), thus trying to find a middle ground between UOFO and RTRL. The correctness of the proposed algorithm is analyzed but only informal theoretical arguments are given for the improvement in terms of variance wrt UORO. Still, the experiments on the synthetic copy/paste task and on the Penn tree bank show that KF-RTRL performs better than UORO in terms of convergence rate of the training algorithm. OPINION: The contribution is interesting and the idea to replace the restrictive rank one approximation of UORO by a more expressive factorization while still achieving better computational complexity than RTRL is worthwhile. Regarding clarity the paper could be first improved by a careful proof reading and polishing. The exposition of the results could also be improved: it is for example clumsy to defer the definition of the class of RNN for which Theorem 1 holds to Lemma 1, when Lemma 1 actually has not much to do with Theorem 1 (it only serve the argument that the class of RNN considered is not too restrictive); the actual definition of the class of RNN in Lemma 1 is also difficult to grasp/parse and would necessitate more discussion (see my questions below). I also find the arguments used to claim that the variance introduced by KF-RTRL is smaller than the one of UORO very 'hand-wavy' and I would have appreciated a more formal analysis. On the positive side, the experiment section indeed suggests that KF-RTRL is an attractive alternative to both BPTT and UORO and the code is directly supplied as supplementary material. REMARKS/QUESTIONS: - line 106-108: I don't understand what is meant here, could you clarify? - Algorithm 1: It may be worth precising what the z^k_j are, what SGDopt is, and which norm is used. - Lemma 1 - definition of the RNN class: What do the matrices W^1,...,W^R correspond to? Is it correct that for vanilla RNN with one hidden layer we would have r=1 and W^1 would be the concatenation of the input and recurrent matrices? How is the bias term of the hidden layer handled in this definition? Does this class encompasses multi-layer RNNs? I think more attention should be put on this definition and more discussion/intuition given. - line 178: what are the distributions over u_{t-1} and A_{t-1} mentioned here? - I think RTRL should be included in the comparison for the experiments in Sections 5.1 and 5.2. Is there a reason why it is not the case? MINOR COMMENTS/TYPOS: -line 51: that -> than -line 58: on -> in -line 164: this sentence is a bit misleading, maybe add 'in expectation' somewhere. -line 176: the phrasing is strange, maybe use 'since u_{t-1}' is a row vector' instead. -line 203: $UORO$ -> UORO -line 230: with out -> without -line 236: do you mean 'validation error' instead of 'RNN error'? -line 240: RHN -> RNN -footnote 1 p.6: it is not clear to me how this simplify the implementation. -caption of Fig 1: an -> a -Fig 1 x-axis legend: what is 'data time'? -line 403: Equqtion

Reviewer 3



This work proposes the Kronecker Factored Real-Time Recurrent Learning (KF-RTRL) algorithm, which is inspired from the Unbiased Online Recurrent Optimization algorithm (UORO). RTRL discards backpropagation through time (BPTT) by allowing online updates of the parameters, yet it is impractical because of the requirement of high cost of time and space complexity. To ease these difficulties, both KF-RTRL and UORO approximate the true gradient of RTRL so as to reduce the cost. UORO reduces both run time and memory requirement to O(n2). UORO obtains the unbiased but noisy estimate. The proposed KF-RTRL aims to reduce the variance of the approximation at the cost of O(n3) in run time and O(n2) in memory. With a series of inductive steps, the final estimation is an unbiased approximation and the variance is decreased as the order O(n−1) by introducing the Kronecter factor. Pros: - The paper is well-written and explained. - Rigorous mathematical proof is provided. Cons: - Although the experiments show that KF-RTRL is better than UORO under RHN, the constraint of RNN architecture is still a nonnegligible concern. - It is not clear whether it is appropriate to consider only the cosine angle in the variance analysis. Optimization should consider both direction and magnitude of the gradient.